# Attachment Styles and Well-Being in Adolescents: How Does Emotional Development Affect This Relationship?

**DOI:** 10.3390/ijerph16142554

**Published:** 2019-07-17

**Authors:** Estefanía Mónaco, Konstanze Schoeps, Inmaculada Montoya-Castilla

**Affiliations:** Department of Personality, Assessment and Psychological Treatment, Faculty of Psychology, University of Valencia, 46010 València, Spain

**Keywords:** attachment to parents, emotional competencies, well-being, adolescence

## Abstract

Attachment relationships with parents, as well as emotional competencies, are protective factors against stress and other physical, mental, and relational health symptoms in adolescence. In this paper, we will examine the mediating role of emotional competencies in the relationship between attachment to parents and the well-being of adolescents, taking into account the influence of gender. There were 1276 Spanish adolescents between 12 and 15 years old (*M* = 13.48; *SD* = 1.09). We measured mother and father attachment relationships (trust, communication and alienation), emotional competencies (perceive and understand emotions, label and express emotions, manage and regulate emotions), and adolescent well-being using the indicators: somatic complaints, stress, satisfaction with life and affectivity. Descriptive analyses, Pearson correlations, and a multi-group path analysis were performed. The results indicated that emotional competencies partially mediate the relationship between attachment to parents and well-being variables. Attachment to one’s mother and father, along with emotional competencies, are relevant variables in adolescent well-being. This highlights the importance of understanding the protective factors of well-being in adolescence, a time when levels of well-being are reduced compared to childhood.

## 1. Introduction

Over the last few decades, the study of human well-being has become a field of growing interest in different disciplines [1]. Subjective well-being refers to how people evaluate their lives. These evaluations involve people’s emotional reactions and moods, and judgment about their life satisfaction and fulfillment in certain domains such as social relationships and professional environment [2]. In a systematic review, Diener & Chan [3] conclude that subjective well-being predicts health and longevity. Considering that a happy person enjoys life longer and experiences better health, it seems even more important to studying the concept of subjective well-being and its relationship with other variables.

One of the great challenges of the social sciences of XXI century has been the promotion of children’s and adolescents’ well-being, stressing the protection of their human rights [4]. However, the study of adolescent well-being is still a research field in progress [5] compared to the large number of studies conducted in adult population [1,6]. Stressing that levels of well-being change throughout the life cycle [7] with adolescents reporting lower levels of life satisfaction than other developmental stages [8], the present study focuses on social and emotional predictors of well-being in adolescence.

It is well-known that adolescent well-being is more than only the mere absence of behavioural disorders such as drug-use [9]. In this sense, the need to study what, more specifically, makes adolescents improve their levels of well-being and which factors could positively influence these levels has arisen [10]. The study of family and emotional variables, which will be handled in the present research, helps us to understand how adolescent well-being develops [11,12,13].

Adolescence is a vital stage and is especially sensitive to the appearance of symptoms of emotional discomfort and instability, given the need to face new challenges and vital changes that generate stress [14]. At the same time, it is a period of flexibility in which the adolescent is open and permeable to new learning, from global knowledge to concrete skills and competencies, such as the ability to manage one’s own emotions [15].

From early studies, parenting literature have analyzed the relation between parent–infant relationships and children’s outcomes [16,17]. Among the social-emotional factors associated with adolescent well-being, parental attachment relationships stand out [18,19]. In addition, a parent–infant relationship is built on trust, communication and lack of alienation, which indicates secure attachment. Such dimensions are conceptually related to the classic parenting factors: warmth and communication [20,21]. Thus, the emotional bonds established between a baby and its primary caregivers during early childhood might be influencing the future mental scheme teenagers form about themselves and the world around them [22,23]. Children whose primary caregivers, usually the mother and the father, were sensitive and responded appropriately to their needs, establish a bond of secure parental attachment and, as a consequence, foster a basic attitude of trust towards others [24]. From adolescence to mid-life, this could translate into more fluid and positive interpersonal relationships, thanks to more effective emotional and interpersonal skills [23,25].

Adolescents with insecure attachment to parents are more likely to engage in risky behaviours, present behavioural problems, and experience difficulties with emotional regulation, such as impulsivity [26,27,28]. Conversely, adolescents who develop a secure attachment relationship with both their parents report greater satisfaction with life and greater positive affect, less stress, stronger self-esteem, and more interpersonal skills [18,27,29,30]. In addition, attachment security is also associated with emotional competence, i.e., a greater ability to perceive, label, express and regulate their emotions [28]. Most previous studies examined only attachment to the mother [31], or did not separate maternal and paternal attachment [32], neglecting the important role of the fathers [33]. Therefore, in this study, we explored attachment to the mother and the father separately, considering that they are equally and strongly related to adolescents’ mental health and well-being [27].

Emotional competencies are defined as the abilities that the individual develops to manage his or her own and other people’s emotions to function properly in a given social context [34]. The ability to understand one’s own emotional states, to express them adequately and to regulate their intensity has been consistently associated with different indicators of subjective well-being [35] and physical well-being, for instance less somatic complaints and lower levels of perceived stress [36,37]. More specifically, somatic complaints are unpleasant body-related perceptions such as stomachache and headache, which are related to emotional dysfunction [38]. Perceived stress relates to a person’s feeling that the demands in their life exceed their capacity to cope effectively [39].

Some research suggests that emotional competencies may function as a mediating variable between attachment to parents and well-being in adulthood [30,40]. This mediation assumes that the development of emotional competencies may buffer the negative effects of maintaining insecure attachment relationships. Furthermore, adolescents who are able to establish secure attachment bonds based on trust and communication, feel more comfortable to talk about their feelings, and therefore understand and know how to cope with them [41]. Thus, emotional abilities could be a protective factor of well-being, although further study is needed, especially in the adolescent population [42].

Lastly, it is important to consider the influence of sociodemographic variables, such as age and sex, when examining the relationship among attachment, emotional competencies and well-being in adolescence [43,44]. In terms of age, the relationship with parents seems to lose relevance in adolescence compared to childhood, with relationships with a peer group being a priority [25,45]. At the same time, as previously mentioned, levels of well-being are reduced with respect to children [46,47]. In terms of sex, there seem to be differences between girls and boys in the understanding, expression and managing of their emotions [48]. Although there is no consensus, the literature suggests that girls are more likely to perceive and express their emotions accurately, while boys develop strategies to regulate their negative emotions more efficiently [49].

Based on existing research [3,6,42], subjective well-being was studied through satisfaction with life and both positive and negative affect, and physical well-being was studied through stress and somatic complaints. Following the procedure recommended by Baron and Kenny [50], the preconditions for mediation analysis are available in the literature: previous studies provide empirical evidence that the predictor variable (parental attachment) appears to be significantly related to the outcome variable (well-being) [22,26]; at the same time, the mediator variable (emotional competence) is significantly related to the predictor [23,24] and outcome variables [29,31]. Furthermore, this study is focused on sex and age-specific differences due to the fact that subjective well-being decreases from early to middle adolescence, with girls presenting lower levels than boys [51]. It is also important to consider that some authors suggest different levels of vulnerability depending on the age of adolescents, being greater in late adolescence than in early adolescence [44]. Maintaining secure attachment bonds and the development of emotional skills and competence at this increasingly vulnerable age is generally hypothesized to be a good predictor of one’s sense of well-being and health [52,53,54,55,56,57].

Given the above, the objective of the present study was to provide additional evidence of the relationship between parental attachment and the well-being of adolescents, considering the mediating role of emotional competencies and taking into account age and gender differences. Based on existing research, the following hypotheses were tested for whether (1) emotional competencies play a mediating role in the relationship between attachment to the father and mother and well-being, and (2) whether there are possible differences in sex and age in this interaction.

## 2. Materials and Methods

### 2.1. Participants

The study involved 1276 students from the 1st to the 4th grade of compulsory secondary education at 10 public, private and subsidized high schools located in the Valencian Community (Spain). Participants’ age ranged from 12 to 15 years (*M* = 13.48; *SD* = 1.09), which is 1st to 4th grade of compulsory high school. Subsamples were distinguished in order to examine potential sex and age-specific differences: girls (*N* = 689, *M* = 12.76, *SD* = 0.76) versus boys (*N* = 587, *M* = 12.83, *SD* = 0.84); early adolescents between 12 and 13 years (*N* = 649, *Mage* = 12.53, *SD* = 0.50) versus late adolescents between 14 and 15 years (*N* = 627, *M* = 14.46, *SD* = 0.50), considering differences in vulnerability according to age.

Regarding parents’ educational level, the distribution was as follows: university degree (31%), high school (32%), primary studies (32%) and no studies (5%). Most of the adolescents lived with both parents at home (78%). Otherwise, in the case where teenagers did not live together with both of them, the reasons varied: 87% divorce, 6% father’s death, 4% mother’s death, and 3% other causes, such as work outside the home or migration of one of the parents. With regard to siblings, 24% were an only child, 63% had two siblings, 16% three siblings and 4% four or more siblings.

### 2.2. Instruments

The instruments used in this study are well-established self-report measures that have been adapted and validated for the Spanish adolescent population.

*Parental attachment.* Parental attachment was measured with the Inventory of Parents and Peers Attachment (IPPA) [58,59]. This instrument, based on Bowlby’s [60] attachment theory, assesses adolescents’ perceptions of relationships with their parents and peers at the affective and cognitive dimension. The instrument assesses attachment to mother, attachment to father and attachment to peers separately. Each measure comprises 25 items with a Likert scale from 1 to 5 (1= Never or almost never; 5= Always or almost always). In the present work, only the scales of attachment to the mother and father were used. Depending on the factor structure of this instrument [61], each attachment scale comprises three dimensions: trust (perception of mutual trust and respect of desires and needs by the parent towards the adolescent; e.g., “My mother/father respects my feelings”; α = 0.87 for mother and α = 0.90 for father), communication (response capacity and verbal communication of the parent regarding the emotional states of the adolescent; e.g., “My mother/father encourages me to talk about my difficulties”; α = 0.84 for mother and α = 0.82 for father) and alienation (feelings of social isolation, anger and detachment towards the parents, even though there is a need to approach them; e.g., “I get easily angry with my mother/father”; α = 0.66 for mother and α = 0.65 for father). The total scale was calculated combining these three dimensions with higher scores indicating more security in the attachment relationship (Total score = Trust + Communication − Alienation). The total scales showed good reliability in the sample of this study (α = 0.90 for attachment to mother and α = 0.91 for attachment to father).

*Emotional competencies.* Emotional competencies were measured using the Emotional Skills and Competence Questionnaire. The Spanish version was reduced to 21 items (ESCQ-21) [62], with a Likert scale of 6 points (1 = Never; 6 = Always). This questionnaire evaluates emotional intelligence from a competency or ability perspective and has three scales: Perception and Understanding (α = 0.84; e.g., “I can differentiate whether my friends are sad or disappointed”), Labelling and Expression (α = 0.90; e.g., “I get to express my feelings in words”) and Management and Regulation (α = 0.78; e.g., “I manage to stay in a good mood, even if something bad happens”). The questionnaire has shown good validity and reliability [50].

*Somatic complaints.* Somatic complaints were assessed using the List of Somatic Complaints (SCL) [63]. The instrument comprises 11 items with a three-point Likert scale (1= Never; 3= Often; e.g., “I have a stomach ache”). Participants indicate how often they experience somatic symptoms such as stomach pain, tiredness, or back pain. The list has shown to be valid and reliable [32], which was also the case in this study (α = 0.80).

*Stress.* Stress was measured using the Perceived Stress Scale (PSS-4) [64,65], evaluating the degree to which situations in the last month are assessed as unpredictable and out of control. The scale comprises four items structured on a Likert scale from 0 to 4 (0 = Never; 4 = Always; e.g., “I felt the difficulties piled up without being able to solve them”). The scale has shown to be reliable and valid [53]. In this study, reliability was also acceptable (α = 0.67).

*Satisfaction with life.* Satisfaction with life was assessed using the Life Satisfaction Scale (SWLS) [66,67]. This short scale assesses people’s satisfaction with their living conditions; it comprises five items, with a Likert scale from 1 to 7 (1 = Completely disagree; 7 = Completely agree; e.g., “Conditions of my life are excellent”). This scale shows good psychometric properties [55]; also in this study (α = 0.85).

*Affectivity.* Affectivity was measured using the Scale of Positive and Negative Experiences (SPANE) [68]. The instrument comprises 12 items, six of which refer to positive experiences and feelings (e.g. “In the last 4 weeks I’ve had happy feelings”) and six of which refer to negative or worrying experiences (e.g. “In the last four weeks I’ve had sad feelings”). Participants are asked to rate how often they have experienced positive and negative feelings over the past 4 weeks on a five-point scale (1 = Never, 5 = Always). Both scales obtained good reliability indices: Positive Affect scale (α = 0.87) and the Negative Affect scale (α = 0.81).

### 2.3. Procedure

First, the evaluation survey for adolescents was drawn up and approved, together with the list of schools, by the Department of Education of the Valencian Community and by the ethical committee of the University of Valencia (H1385330676977). Second, those schools that were interested in taking part in the study were contacted. Prior to the data collection, parents were asked to sign a written and informed consent for their children’s participation in the research. If the parents did not sign the consent form, students were excluded from the study. Lastly, student assessments were conducted in a 50-minute session during school hours and in the presence of teachers. In carrying out this work, the World Medical Assembly ethical standards were respected in the Helsinki Declaration.

### 2.4. Data Analysis

Basic descriptive statistics and Pearson bivariate correlations were carried out to estimate the relationship between variables.

To test our hypotheses regarding gender and age-specific mediation, we conducted a multi-group path analysis (MGPA) to express the relationship between variables in the path model using regression equations and assessing mediation within two subgroups (age and gender) with the causal step procedure [50,68,69]. Structural Equation Modeling (SEM) analysis with observed variables was conducted to predict the pathways from father and mother attachment, which are estimated through trust, communication and alienation, to somatic complaints, perceived stress, life satisfaction, positive affect, and negative affect, acting emotional competence as a mediator, which is estimated through perceiving and understanding emotions, expressing and labelling emotions, and managing and regulating emotions. Total effects, direct effects and indirect effects were estimated, constructing bootstrap confidence intervals (CI) around the estimates to assess the effects of mediators [45,62].

We conducted a stepwise multigroup analysis to evaluate potential differences among two subgroups, girls and boys and students aged 12–13 years old (first cycle of secondary school) and 14–15 years old (second cycle of secondary school). In the first step, we applied the unrestricted model, with all parameters estimated freely (baseline model). In the second step, a semi-restricted model assuming equal factor loadings, free thresholds, and free regression coefficients was tested for both subgroups. In a third step, we applied the fully restricted model, assuming equal factor loadings, equal thresholds, and equal regression coefficients across gender and age groups. The semi-restricted and fully restricted models were compared using the χ^2^ difference test [70].

All analyses were conducted using the statistical software Mplus version 7.0 [71] and maximum likelihood estimation (MLR) with robust standard errors and chi-square values. In addition, confidence intervals around the estimates have been constructed to assess the effects of mediators [72,73], which diminish bias caused by the non-normality in the sampling distribution of indirect effects [74].

Model fit was estimated in Mplus using four main fit indices for the model fit as recommended by Hu and Bentler [75]: chi-square test of model fit (χ^2^), root mean square error of approximation (RMSEA), comparative fit index (CFI), and standardized root mean square residuals (SRMR). To account for missing data, the models were estimated with full information maximum likelihood (FIML).

## 3. Results

### 3.1. Descriptive Statistics and Relationships among Variables

Descriptive statistics (range, means, standard deviation, skewness and curtosis) are presented in Table 1. Skewness values below 2 and Kurtosis values below 7 indicate normal distribution, thus the variables of this study can be handled as normal distribution [76].

Bivariate correlations have been performed to study the relationship between variables. Regarding the relationship between attachment and emotional competencies (Table 2), it has been shown that the total score of attachment security to mother and father is significantly and positively related to emotional competencies. Similarly, all dimensions of parental attachment (trust, communication, alienation) are significantly related to welfare variables. Specifically, attachment to both parents is positively and meaningfully related to life satisfaction and positive affect, and meaningfully and negatively to somatic complaints, stress and negative affect. The highest correlations are with life satisfaction and stress.

In terms of emotional competencies, it has been shown that the perception and understanding of emotions is significantly negatively related to stress and significantly positively related to life satisfaction and positive affect. It does not correlate significantly with somatic complaints or negative affect. The scales of expression and labeling, along with handling and regulation, correlate significantly negatively with somatic complaints, stress and negative affect, and positively with life satisfaction and positive affect. The strongest correlations are observed between emotional management and regulation, and satisfaction with life and positive affect.

### 3.2. Mediating Role of Emotional Competence

We conducted a multi-group path analysis (MGPA) expressing mediation in terms of direct, indirect, and total effects and estimated how paths that constitute these effects vary across gender and age groups. The model included direct paths (1) from mother and father attachment (trust, communication and alienation) to perceived emotional competence (perceive, express and manage emotions) and self-report measures of well-being (life satisfaction, positive and negative affect, somatic complains and perceived stress), along with (2) emotional competence to well-being. In addition, the model comprised an indirect path from parent attachment to well-being through emotional competence (Figure 1). Model fit indices indicated a good fit of the data: χ^2^ (49) = 305.26, *p* < 0.001, χ^2^/gl = 6.23, RMSEA = 0.06 [0.05–0.07], CFI = 0.96, TLI = 0.90, SRMR = 0.05.

*Total, direct and indirect effects*. In this mediation model, the associations between the emotional competence of perceiving, expressing and managing emotions and measures of well-being (such as life satisfaction, positive and negative affect, somatic complains and perceived stress) were highly significant (Table 3). In addition, the direct effects of mother and father attachment on such emotional competence were significant, and the direct paths of parent attachment and life satisfaction, positive and negative affect, and perceived stress were significant, indicating partial mediation, even though the direct effect of mother attachment on somatic complains was not significant, indicating full mediation. The indirect effects of mother and father attachment mediated by emotional competence were found to be significant. Overall, 41% of the variance of life satisfaction (*R*^2^ = 0.41), 33% of the variance of positive affect (*R*^2^ = 0.33), 10% of the variance of negative affect (*R*^2^ = 0.10), 16% of the variance of somatic complaints (*R*^2^ = 0.16) and 32% of the variance of perceived stress (*R*^2^ = 0.32) were explained through the combined effects of this mediation model, taking into account shared variance among variables (63).

### 3.3. Gender and Age Differences

For the previously estimated model, three multi-group path analyses for each subgroup (unrestricted model, semi-restricted model and fully-restricted model) were carried out to identify potential sex and age-specific differences in the association between parent attachment and self-report measures of well-being mediated by perceived emotional competence among adolescents (Table 4).

*Differences between gender groups.* To evaluate gender differences, in the first step, an unrestricted baseline model was established estimating all parameters freely. In a second step, a less restricted model with factor loadings held equal between the two groups was used, whereas the thresholds and regression coefficients were allowed to be free and were computed. This less-restricted model showed a favourable fit. In a third step, a more restricted model was computed, in which the factor loadings, thresholds and regression coefficients between the groups were held equal. The indices showed an adequate fit for this more-restricted model. Using a χ^2^-difference test, the more-restricted model was compared to the less-restricted model. At first, the chi-square differences test between the two models was significant, Satorra-Bentler χ^2^ (31) = 109.93, *p* < 0.001. A second fully-restricted model, assuming different regression coefficients for five of the direct paths from parental attachment to emotional competence and to the different well-being outcomes, was compared to the semi-restricted model (Figure 2). The test did not reach significance (χ^2^ (19) = 24.36, *p* = 0.14), which implies that the more restricted multi-group model fits the data no worse than the less restricted model. Thus, female and male adolescents did not differ considerably in the impact of the examined interplay.

*Differences between age groups.* The same stepwise analysis was used to evaluate age differences. Thus, after establishing a freely-estimated baseline model, a semi-restricted model with equal factor loadings was computed. This less-restricted model showed a favourable fit. Subsequently, a more constrained model was estimated, in which the factor loadings, thresholds and regression coefficients between the groups were held equal. The indices showed an adequate fit for this fully-restricted model. The initial chi-square differences test between the two models was significant, Satorra- Bentler χ^2^ (31) = 81.93, *p* < 0.001. In a second attempt, a fully-restricted model assuming different regression coefficients for two of the direct paths from parental attachment to emotional competence and to the different well-being outcomes, was compared to the semi-restricted model. The test did not reach significance this time (*χ*^2^ (27) = 32.78, *p* = 0.23), which implies that the more-restricted multi-group model fits the data better than the less-restricted model (Figure 3). Thus, adolescents aged 12–13 and 14–15 years fairly resembled each other in the examined interplay.

## 4. Discussion

Research has shown that the quality of adolescents’ social relationships plays a significant role in their well-being [18,29]. Although in adolescence the relationship with the peer group acquires greater relevance than in childhood, parent-child relationships continue to be a primary source of well-being for the adolescent [25]. One of the variables which influences the development of emotional competencies is parental attachment, in fact, adolescents who maintain a secure attachment to their parents are more capable to perceive, name, express and regulate their emotions [77]. With all this, the objective of this study was to understand the influence of attachment to mother and father on the well-being of adolescents, contemplating the mediating role of emotional competencies in this relationship, given that these could be a damping factor in the development of discomfort in the face of unhealthy paternal-filial attachment bonds [30].

The results obtained indicate, on one hand, that having a bond of secure attachment to the mother and to the father is positively related to satisfaction in the different vital areas of the adolescent. Secure attachment also relates to increased positive affectivity, i.e., positive experiences, vitality, interest, and positive mood. On the other hand, secure parental attachment is negatively related to negative moods, i.e., negative experiences and unpleasant moods (e.g., anger, disgust, guilt, fear, or nervousness); it is also associated with lower perception of stress and fewer somatic complaints, such as tiredness, body discomfort or nonspecific pain. Likewise, adolescents with secure parental attachment relationships are more emotionally competent and have a greater ability to perceive, express, name and manage their own emotions, as well as those of others. These findings are in line with what has been proposed in the previous literature in the adult and adolescent population [18,29,36,40].

Well-developed emotional competencies consist of perceiving and understanding one’s own and other people’s emotions, naming and adequately expressing what we are feeling, and managing and regulating both positive and negative emotions [34]. Emotional competencies relate positively to life satisfaction, positive experiences and moods, and relate negatively to the perception of stress e.g., [78]. The ability to label, express, manage and regulate emotions is negatively related to somatic complaints and negative experiences and moods [14,36,37]. However, the ability to perceive and understand emotions is not related to a lower level of somatic complaints or negative affect; this relation could be due to the fact that excessive attention and an understanding of negative emotions, if it is not accompanied by a strong capacity for emotional regulation, does not benefit well-being and health [78].

In line with the first hypothesis, the results obtained confirm that emotional competencies have a mediating role between parental attachment and the well-being of adolescents. This result means that parental attachment indirectly influences the well-being of the adolescent through emotional competencies. However, we consider it important to point out that attachment to parents is a variable that continues to have a direct influence on the well-being of adolescents.

As an exception, attachment to the mother ceases to be an influential factor in the development of somatic complaints when the role of emotional competencies is taken into account. In other words, if the adolescent knows how to manage his or her emotions properly, having an unsatisfactory relationship with the mother is no longer a relevant factor in the presence of physical discomfort. These results provide previous studies [30,41,79] with a better understanding of the mediating relationship between the variables worked on.

According to the second hypothesis, differences would be expected depending on the age and sex of the adolescents. In the age results, few differences were found between the younger (12–13 years) and older (14–15 years) age groups. It was only observed that, in the case of younger adolescents, attachment to the father does not significantly influence positive affect when considering the mediating role of emotional competencies. This means that the level of positive experiences and emotions of 12-to-13-year-old adolescents depends on their ability to manage emotions, regardless of the quality of their parental bonding. However, in 14-to-15-year-olds, this parental bonding is just as important as their emotional competencies. These results run counter to the literature review, which states that the importance of parental bonding decreases progressively as the age of adolescents increases [45].

Lastly, two significant changes were observed when taking into account the influence of sex in the proposed mediation. First, attachment to the mother ceased to directly influence somatic complaints in the case of males, while it continued to influence the case of females. Second, attachment to the father ceased to have a direct influence on positive affect in the case of men, but not in the case of women.

Furthermore, we note that the association between parental attachment and emotional competencies is weaker in males than in females. These results would indicate that in the case of women, attachment to both fathers and mothers is more relevant than in the case of men, since they continue to be influential factors in the well-being of girls, even though girls have a good capacity to manage their emotions [43,49].

In conclusion, the novelty that our results bring to the existing literature is the mediating function of emotional competencies between attachment to parents and the well-being of adolescents and how this interaction varies according to sex and age. Our study suggests that adequate training in emotional competencies may increase life satisfaction and positive affect in adolescents [67]; it may also reduce stress, somatic complaints, and negative affect, even in those teens with a weak attachment to their parents [52].

Even when considering the importance of emphasizing the development of the emotional competencies of adolescents, the results obtained lead us to emphasize the importance of satisfactory family relationships in adolescence. Although at this stage young people become more autonomous and focus on their peer group [14], attachment to parents remains an equally important factor in well-being, especially for girls.

This research makes an important contribution to well-being in the adolescent stage, since it is a population with particular characteristics and needs, different from those existing in other evolutionary stages such as childhood or adulthood [15]. Moreover, we consider a strength of our work to be that it was carried out with Spanish adolescents, since in addition to the fact that there are few studies using a Spanish-speaking population, the results obtained in other countries should not be generalized due to possible cultural and contextual differences in the psychological variables studied [80].

It is important to recognize that our study is not without limitations. One of our main limitations is in having used self-report questionnaires to evaluate the variables studied. Thus, we keep in mind, that at all times, we are working with the subjective perception that adolescents have of their attachment relationships, their emotional competencies and their well-being. This subjective perception may not necessarily coincide with objective behaviour, especially in the case of emotional competencies [42]. In addition, in future investigations, it would be interesting to enlarge and diversify the sample used, including longitudinal measurements taken at different times. This would make it possible to conduct causality studies and to more safely locate the variables that influence adolescent well-being [24].

To conclude, two future lines of work for professionals in psychology, education and health can be deduced from the present work. First, it is important to promote emotional education programmes that develop the emotional competencies of adolescents, and the introduction of this type of training either in formal education or extracurricular education [66]. Second, the need to provide parents with a variety of parenting resources, i.e., the knowledge and skills needed to bond safely with their children from early childhood and to adequately address the challenges that arise in their developmental maturation through adolescence [45]. In line with other authors [27], we consider that these strategies would be an efficient means of promoting well-being and preventing emotional and behavioural problems in adolescents.

## 5. Conclusions

One of the great challenges of the social sciences of XXI century has been the promotion of children’s and adolescents’ well-being, considering its impact on health [3,4]. However, the study of adolescent well-being is still in progress [5]. Stressing that levels of well-being change throughout the life cycle [7] with adolescents reporting lower levels of life satisfaction than other developmental stages [8], the present study focuses on social and emotional predictors of well-being in adolescence.

The findings of the present study confirm that emotional competencies (ability to label, express, manage and regulate emotions) have a mediating role in the relationship between parental attachment and well-being of adolescents, measured by somatic complaints, perceived stress, life satisfaction and affectivity. In early adolescence (12–13 years), attachment to the father does not significantly influence positive affect when considering the mediating role of emotional competencies, compared to older adolescents (14–15 years). Furthermore, the association between parental attachment and emotional competencies is weaker in males than in females.

These findings suggest that the development of emotional competencies may increase life satisfaction and positive affect in adolescents; it may also reduce perceived stress, somatic complaints, and negative affect, even in those teenagers with a weak attachment to their parents.

## Figures and Tables

**Figure 1 ijerph-16-02554-f001:**
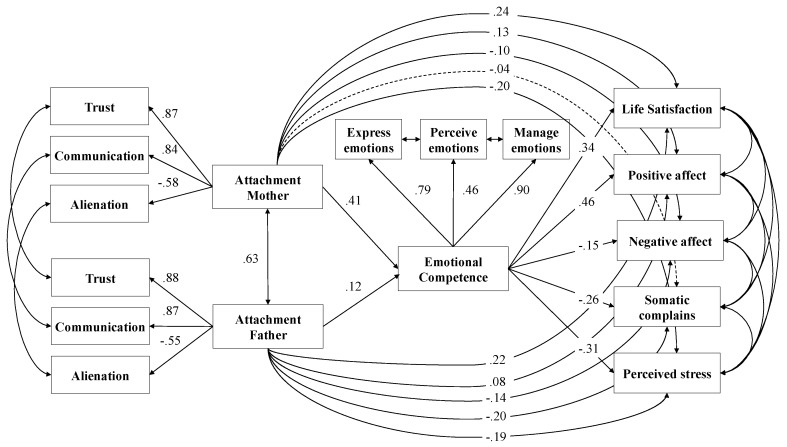
Path model: Interplay of mother and father attachment with self-reported measures of well-being mediated by perceived emotional competence. *Note:* Significant effects shown as standardized coefficients (β); continuous pathways are significant at *p* < 0.01, dotted pathways are not significant; factor loadings are standardized.

**Figure 2 ijerph-16-02554-f002:**
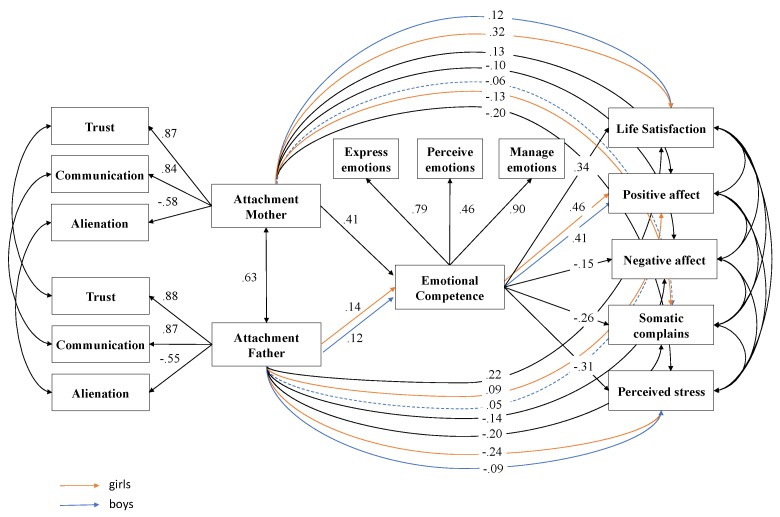
Multi-group path model: Sex differences. *Note:* Significant effects shown as standardized coefficients (β); continuous pathways are significant at *p* < 0.01; dotted pathways are not significant; factor loadings are standardized.

**Figure 3 ijerph-16-02554-f003:**
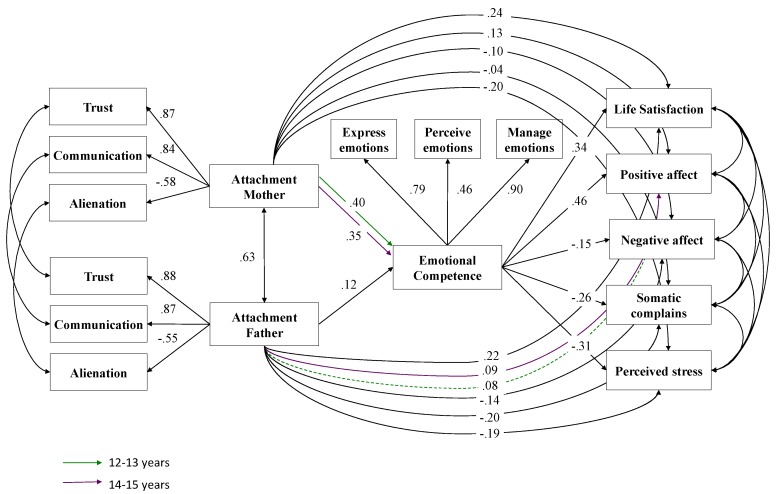
Multi-group path model: Age differences. *Note:* Significant effects shown as standardized coefficients (β); bold pathways are significant at *p* < 0.01; dotted pathways are not significant; factor loadings are standardized.

**Table 1 ijerph-16-02554-t001:** Means (*M*), Standard Deviations (*SD*), Range, Skewness and Kurtosis.

Measures	Rank	*M* (*SD*)	Skewness	Kurtosis
1. Mother attachment	83	62.34 (14.83)	−0.70	0.15
1.1. Trust mother	48	41.99 (6.30)	−1.55	3.63
1.2. Communication mother	41	34.26 (6.87)	−0.63	0.12
1.3. Alienation mother	28	14.03 (4.55)	0.25	−0.39
2. Father attachment	84	57.73 (16.51)	−0.83	0.62
2.1. Trust Father	48	40.20 (7.79)	−1.51	2.68
2.2. Communication father	88	31.38 (8.10)	−0.19	1.94
2.3. Alienation father	28	14.31 (4.91)	0.41	−0.13
3. Perceive and understand emotions	68	67.08 (10.84)	−0.54	0.74
4. Express and label emotions	65	59.03 (11.62)	−0.48	0.18
5. Manage and regulate emotions	67	73.54 (10.41)	−0.68	0.67
6. Somatic complaints	2	1.47 (0.32)	1.18	1.80
7. Stress	14	8.27 (2.18)	0.36	0.19
8. Life satisfaction	30	25.71 (6.44)	−0.68	−0.12
9. Positive affect	25	22.40 (4.88)	−0.46	−0.24
10. Negative affect	2	12.93 (4.58)	0.40	0.04

**Table 2 ijerph-16-02554-t002:** Intercorrelations among variables studied.

Measures	2	2.1	2.2	2.3	3	4	5	6	7	8	9	10
1. Mother attachment	0.71 **	0.55 **	0.63 **	−0.40 **	0.21 **	0.36 **	0.38 **	−0.29 **	−0.46 **	0.52 **	0.38 **	−0.29 **
1.1. Trust mother	0.60 **	0.57 **	0.49 **	−0.30 **	0.18 **	0.27 **	0.32 **	−0.21 **	−0.38 **	0.45 **	0.31 **	−0.21 **
1.2. Communication mother	0.66 **	0.45 **	0.67 **	−0.33 **	0.24 **	0.37 **	0.39 **	−0.20 **	−0.35 **	0.44 **	0.33 **	−0.19 **
1.3. Alienation mother	−0.50 **	−0.37 **	−0.39 **	0.41 **	-0.07 *	−0.27 **	−0.26 **	0.35 **	0.48 **	−0.43 **	−0.32 **	0.35 **
2. Attachment father		0.90 **	0.90 **	−0.69 **	0.18 **	0.32 **	0.36 **	−0.31 **	−0.45 **	0.57 **	0.35 **	−0.28 **
2.1. Trust father			0.76	−0.46 **	0.14 **	0.21 **	0.27 **	−0.27 **	−0.36 **	0.43 **	0.27 **	−0.22 **
2.2. Communication father				−0.43 **	0.22 **	0.32 **	0.33 **	−0.25 **	−0.35 **	0.44 **	0.28 **	−0.20 **
2.3. Alienation father					−0.07	−0.20 **	−0.23 **	0.27 **	0.39 **	−0.40 **	−0.27 **	0.29 **
3. Perceive and understand emotions						0.59 **	0.66 **	−0.02	−0.22 **	0.27 **	0.25 **	−0.02
4. Express and label emotions							0.69 **	−0.26 **	−0.39 **	0.42 **	0.41 **	−0.20 **
5. Manage and regulation emotions								−0.26 **	−0.41 **	0.48 **	0.49 **	−0.19 **
6. Somatic complaints									0.43 **	−0.37 **	−0.37 **	0.41 **
7. Stress										−0.55 **	−0.43 **	0.41 **
8. Life satisfaction											0.49 **	−0.33 **
9. Positive affect												−0.36 **
10. Negative affect												

Note: * *p* < 0.05; ** *p* < 0.01.

**Table 3 ijerph-16-02554-t003:** Coefficients of total, direct, and indirect effects.

Paths	Total Effect			Direct Effect			Indirect Effect		
	c	SE	95% CI	c’	SE	95% CI	ab	SE	95% CI
Attachment Father ➔ Emotional Competence ➔ Life Satisfaction	0.26 ***	0.04	(0.17, 0.34)	0.22 ***	0.04	(0.14, 29)	0.04 **	0.01	(0.01, 0.07)
Attachment Mother ➔ Emotional Competence ➔ Life Satisfaction	0.37 ***	0.04	(0.29, 0.46)	0.24 ***	0.04	(0.15, 0.32)	0.14 ***	0.02	(0.10, 0.18)
Attachment Father ➔ Emotional Competence ➔ Positive Affect	0.14 ***	0.04	(0.06, 0.21)	0.08 **	0.03	(0.01, 0.15)	0.05 **	0.02	(0.02, 0.09)
Attachment Mother ➔ Emotional Competence ➔ Positive Affect	0.32 ***	0.04	(0.24, 0.40)	0.13 ***	0.04	(0.05, 0.21)	0.19 ***	0.02	(0.14, 0.23)
Attachment Father ➔ Emotional Competence ➔ Negative Affect	−0.16 ***	0.04	(−0.24, −0.08)	−0.14 ***	0.04	(−0.22, −0.07)	−0.02 **	0.01	(−0.03, −0.01)
Attachment Mother ➔ Emotional Competence ➔ Negative Affect	−0.16 ***	0.07	(−0.24, −0.08)	−0.10 *	0.04	(−0.18,−0.01)	−0.06 ***	0.02	(−0.10,−0.03)
Attachment Father ➔ Emotional Competence ➔ Somatic Complaints	−0.23 ***	0.04	(−0.32, −0.14)	−0.20 ***	0.04	(−0.28, −0.11)	−0.03 **	0.01	(−0.05, −0.01)
Attachment Mother ➔ Emotional Competence ➔ Somatic Complaints	−0.14 ***	0.04	(−0.23, −0.06)	−0.04	0.05	(−0.13, 0.05)	−0.11 ***	0.02	(−0.14, −0.07)
Attachment Father ➔ Emotional Competence ➔ Perceived Stress	−0.23 ***	0.04	(−0.31, −0.15)	−0.19 ***	0.04	(−0.26, −0.11)	−0.04 **	0.01	(−0.06, −0.01)
Attachment Mother ➔ Emotional Competence ➔ Perceived Stress	−0.32 ***	0.04	(−0.40, −0.25)	−0.20 ***	0.04	(−0.28, −0.11)	−0.13 ***	0.02	(−0.17, −0.09)

Note: c, c’, ab = Estimators of total, direct and indirect effects. SE = Standard Error. 95 % CI = 95% bootstraps Confidence Intervals; *** *p* < 0.001. ** *p* < 0.01. * *p* < 0.05.

**Table 4 ijerph-16-02554-t004:** Fit Indices for Multigroup SEM and χ^2^-Difference Test Results.

Model	*df*	χ^2^	*p*	CFI	RMSEA	90% CI	SRMR
**Gender subgroup:**							
**Girls**	51	192.970	0.00	0.95	0.06	(0.05–0.07)	0.05
Boys	51	205.728	0.00	0.95	0.07	(0.06–0.08)	0.06
Model A	102	398.647	0.00	0.95	0.06	(0.06–0.07)	0.05
Model B	114	488.325	0.00	0.94	0.07	(0.06–0.07)	0.08
Model B2	112	437.250	0.00	0.95	0.06	(0.06–0.07)	0.07
Model C	143	546.697	0.00	0.93	0.06	(0.06–0.07)	0.09
Model C2	131	457.751	0.00	0.95	0.06	(0.06–0.07)	0.08
Model compared to each other					Δ χ^2^	*p*	Δ *df*
Model B compared to Model A					86.44	0.00	12
Model B2 compared to Model A					34.59	0.08	24
Model C compared to Model B2					109.93	0.00	31
Model C2 compared to Model B2					24.36	0.14	19
Age subgroup:							
12–13 years	51	167.380	0.00	0.96	0.06	(0.05–0.07)	0.05
14–15 years	51	283.610	0.00	0.93	0.08	(0.07–0.09)	0.07
Model A	102	453.482	0.00	0.94	0.07	(0.06–0.08)	0.06
Model B	112	484.214	0.00	0.94	0.07	(0.06–0.08)	0.06
Model B2	123	486.275	0.00	0.94	0.07	(0.06–0.07)	0.06
Model C	145	563.674	0.00	0.93	0.06	(0.06–0.07)	0.07
Model C2	141	510.458	0.00	0.94	0.06	(0.06–0.07)	0.07
Model compared to each other					Δ χ^2^	*p*	Δ *df*
**Model B compared to Model A**					57.36	0.00	12
**Model B2 compared to Model A**					32.29	0.06	21
**Model C compared to Model B2**					81.93	0.00	31
**Model C2 compared to Model B2**					32.78	0.23	27

*Note.* Model A = unrestricted baseline model (all parameters free); Model B = semi-restricted model (free regression coefficients across sex); Model B2 = adjusted semi-restricted model (free intercepts); Model C = fully restricted model (equality of regression coefficients); Model C2 = adjusted fully restricted model (free regression coefficients). *df* = degrees of freedom. χ^2^ = chi-square test of model fit. *p* = *p*-value. CFI = Comparative Fit Index. RMSEA = Root Mean Square Error Of Approximation. 90% CI = 90% Confidence Interval. SRMR (Standardized Root Mean Square Residual). Δ χ^2^ = Satorra-Bentler Scaled Chi Square.

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
