# Peer review of "Attachment Styles and Well-Being in Adolescents: How Does Emotional Development Affect This Relationship?"

_ijerph, 2019, doi:10.3390/ijerph16142554_

Round 1

Reviewer 1 Report

Review “Attachment styles and well-being in adolescents: how does emotional 
development affect this relationship?”

Revise and Resubmit with Major Revisions*

General comment:

The main strength is in the mediation model but detailed data about the correlations are missing, therefore the respect of the assumptions for the mediation are impossible to check. Overall, the paper is good written and balanced, but its potential is covered by the lack of information and some incongruence among the description of the measures, the data analyses and the results reported.

Comments to the Author

Thanks for the opportunity of revising your manuscript entitled “Attachment styles and well-being in adolescents: how does emotional development affect this relationship?”. The manuscript has the strength to take into account the emotional development in mediate the quality of the relationships with the parents during adolescence, a developmental stage still poor studied from an attachment perspective. Another strength is the large sample size (N = 1276). In spite of these considerable strengths, some needful results are reported as “significant” but without reporting data to check the assumptions of the mediation model, therefore readers cannot evaluate the relevance of the results. In addition, in my opinion there are some weaknesses need to be addressed before the publication:

ABSTRACT

 Line 20 “TD=1.09” did authors mean SD?

INTRODUCTION

p. 2

line 58 to 69, relevance of the study. This part sounds contradictory, because the authors at first outlined that insecure attachment to parents may affect the development of emotional competencies, then they shifted to propose an opposite model where the emotional competencies mediate the negative effect of insecure attachments. Authors should better highlight that not only the security of attachment influence the development of emotional competencies, but also the opposite direction, adding information supporting their study. In sum, in order to highlight the relevance of the study, authors should report studies that found that emotional competencies during adolescence are related to the attachment security, although they are not completely dependent to the attachment security, highlighting the studies reporting mediation effect of emotional competence on attachment and, if possible, emphasizing the controversial findings in literature.

Line 82 uncorrected position. The sentence “In this research, following other studies [30–32], well-being was studied both through negative 80 indicators ─ somatic complaints, stress and negative affect ─ and positive indicators ─ satisfaction 81 with life and positive affect. In addition, following the procedure recommended by Baron and 82 Kenny [33][…] ], the preconditions for mediation analysis are available: the predictor variable (parental 83 attachment) appears to be significantly related to the outcome variable (well-being); at the same 84 time, the mediator variable (emotional competence) is significantly related to the predictor and 85 outcome variables is related to the data analysis and has to be moved in the METHOD section. Specifically, it constitutes a preliminary analysis necessary to proof that the assumptions of the mediation model have been respected, therefore it should be explained in the data-analysis section and it should be briefly detailed in the RESULTS section.

METHOD

p. 3

Participants

Line 95 – 98. May the authors add some information about participants? e.g. family structure (i.e. married and living together parents, separated or divorced parents, etc.), parental level of education, family’s SES, participants disabilities or diagnosis, etc. Are there any participants who have rejected to be involved in the study? Why?

Instruments

line 103 the correct name is Inventory of Parent and Peer Attachment and authors should describe also the total score of security scale, not only the subscales, because it is not clear on what they tested the mediation model (see below, results section).

Line 126 reliability with Cronbach’s alpha .80 is more than acceptable, it is good to excellent value.

Data-analysis

-          Can Pearson correlations be considered as “descriptives”?

Line 155, Missing needful information and data. Authors should first add the preliminary analyses to report the detailed correlations between variables, supporting the preconditions of the mediation model mentioned in the introduction (that is a wrong place to present them). Then they should explain how they investigated their main and first aim, that is the mediation of the emotional competencies between attachment and well-being among adolescents. Indeed, the study's goal is not the mediation of sex and gender that is just an analysis with a control purpose, as they presented it.

RESULTS

Line 182. Missing needful information and data. As above mentioned, here you have the main weakness: authors claimed that the correlations supported the mediation model without reporting data or details on it, those need to be detailed and reported. Without those data, the following results cannot be verified or be considered reliable.

P. 4, line 200. Unclear analyses and incongruence. The table reported correlations with a single measure of attachment each parent, that is (supposed to be) the total score of attachment security, which has been never presented in the description of the IPPA (should be add) in the MEASURES section. Moreover, the purpose suggests that the mediation of emotional competencies is on the total attachment security. However, results focused the subscales of trust, communication and alienation, which has not been mentioned or motivated in the INTRODUCTION section, nor reported in the correlations, but they are reported in the description of the IPPA (should be add) in the MEASURES section as well as in the above-mentioned results of the mediation model.

Authors should add the correlations with all the IPPA scales and subscales, or they have to present a model only with the score of attachment security for each parent, or both.

p.5 FIGURE 1 and 3 are supposed to have significant pathways bold, but they are not bold. Authors should edit the figure or add * and ** to the standardized coefficients to highlight the significance.

DISCUSSION

Overall, authors should revise the English both here and in the other sections, because the contents are clear but the language is sometimes vague, uncertain, and hard to understand in somewhere.

The discussion part could be interesting but the reader cannot appreciate it because of the weakness in the core of the paper, which undermines the reliability of the results from page 4 onwards.

Reviewer 2 Report

Overall nice work. I would have liked more of an explanation for why you chose to break up the sample into the two age groups you did and why you collected data from 12-15 year olds rather than some other age range (12-18 years). 

I would have liked to have seen an example item for each of the measures you used. 

Reviewer 3 Report

Review ijerph-517347

Thank you for your work on this study! You have an enormous sample, which is really great. The fact that you are able to distinguish between fathers/mothers and sons/daughters is also really of interest. However, I think the manuscript needs a lot of work before it would be suitable for publication. I hope that my comments below will help you to do so.

Reference use

I think that the authors should check their references very carefully before resubmitting this article to any journal. Right now, it seems that the references are used incorrectly. I checked some paragraphs on the second page and already found a lot of errors. For example, at line 47 the authors refer to [7] to say something about the importance of parental attachment relationships but that article is not about parental attachment relationships. In addition, in the same line, it is stated that relationships established between a baby and its primary caregivers during early childhood form the future adolescent’s mental scheme about himself and the world around him. However, the quality of attachment between young children and adolescents is not as high as this sentence suggests. I know that Bowbly theorized about this, but it should be made clear that it is just a theoretical idea which is not very strongly supported by evidence (yet). At line 57, source [18] is used as support for the statement in that particular sentence but that source is not about attachment. This errors make me very conspicuous of the reference use in the rest of the paper.

Outline of the introduction

The authors provide a lot of information in the introduction. However, what I miss is:

A clear definition of the main variables in the study (e.g., what is well-being, what entails the attachment relationship between parents and adolescents?). Make sure all the concepts from your method sections are clearly defined in the introduction.  

Why would you expect a relationship between well-being and attachment? What is your theoretical model behind this?

Why would you expect emotional competences to be a mediator?  What is your theoretical model behind this?

Why would you expect differences for girls and boys? And for different age groups?

Would you expect differences between fathers and mothers?

Next to your theoretical ideas: what have empirical studies found so far?

To which knowledge gap would you to contribute? What do we now yet know and why should we know that?

Method section

Seem to entail all information, although a bit more broad description of the population would be helpful to be able to judge how this population reflects the general population of adolescents. In addition, the Cronbach’s alpha for the total attachment scales are given while the authors report about the subscales in the results. What are the Cronbach alpha’s for these subscales?

In addition, in a strict sense you can not test a mediation model when you only have cross-sectional data.

Results

I miss descriptives for each variable.

I’m not familiar enough with the run models to judge whether they did it correctly. However, I don’t understand how the same information (line 248-250) and (lines 264-266) lead to different conclusions.

Discussion

I find it hard to review this section because I have my doubts about the literature used in the introduction and about the inconsequent argumentation in the results section. So I think the article would need major revision first, before I would be able to review this section.

Round 2

Reviewer 1 Report

Thanks for the opportunity of revising your manuscript entitled “Attachment styles and well-being in adolescents: how does emotional development affect this relationship?” and to have addressed all the revisions suggested in a collaborative and satisfying way. Now the paper can be entirely appreciated and the results provide interesting additional information to the existing literature, also well discussed in the discussion section.

Maybe, authors should still highlight more the partial independence of the emotional competencies from the attachment, as their development is influenced by early attachment bonds but also from other variables. Moreover, I suggest to the authors to add the following reference:

Pace C.S., San Martini P., & Zavattini G.C. (2011). The factor structure of the Inventory of Parent
and Peer Attachment (IPPA): a survey of Italian adolescents. Personality and Individual Differences, 51, 83-88. DOI: 10.1016/j.paid.2011.03.006

This is a good paper on the factor structure of the IPPA in its last version (three forms: mother, father, peers)

Author Response

We thank the reviewer for the suggestions made, which we applied to the manuscript to improve its quality.
